# TadA-Bench: A Million-Variant Benchmark for Future-Round Discovery Toward Agentic Protein Engineering

**Jin Gao** [1]   **Juntu Zhao** [1]   **Zirui Zeng** [1]   **Jiaqi Shen** [1]   **Junhao Shi** [1]   **Dukun Zhao** [1]   **Yuming Lu** [1 †]   **Dequan Wang** [1 2 †]

## Abstract

AI for scientific discovery is entering an agentic era, where protein-engineering systems are expected to prioritize future wet-lab experiments rather than merely fit static measurements. We introduce TadA-Bench, a million-variant wet-lab replay benchmark from 31 TadA directed-evolution rounds for future-round discovery toward agentic protein engineering. TadA-Bench preserves the campaign chronology and defines a fixed-data replay task: given earlier experimental rounds, models rank variants that appear only in later rounds. It provides aligned DNA, RNA, and protein views, and uses Seq2Graph, a graph-based label-unification pipeline, to reconcile noisy enrichment measurements into consistent cross-round activity labels. Random-split controls show strong interpolation, but future-round ranking and finite-budget candidate selection are much weaker. Controlled analyses suggest that evolutionary coverage is more informative than local data density, positioning TadA-Bench as a reproducible wet-lab replay substrate for future-round discovery toward agentic protein engineering; the data and code are released on Hugging Face and GitHub.

## 1. Introduction

AI for science is moving toward iterative, agent-like workflows, in which models are expected to participate in scientific decision-making rather than passively fit static datasets. Protein engineering is a representative frontier: an assistant may need to read wet-lab histories, use analysis tools, rank candidate mutations, and help decide what to test next. Such workflows require a data substrate with chronological replayability, exploration scale, and cross-round label consistency. This setting calls for benchmarks that turn real experimental campaigns into reproducible past-to-future discovery problems before agent deployment.

We introduce **TadA-Bench** as a concrete fixed-data implementation of this substrate. As summarized in Figure 1, TadA-Bench preserves chronological replayability by turning 31 rounds of TadA directed evolution into a task where models use earlier variants and measurements to rank later-round variants. It provides exploration scale through a million-variant campaign and supports cross-round label consistency by aligning DNA, RNA, and protein views through **Seq2Graph**, a graph-based pipeline that reconciles noisy enrichment measurements using within-round rankings and cross-round sequence overlaps.

Constructing the benchmark requires turning noisy multi-round sequencing into a stable label space. Selection data contain batch effects, noisy enrichment estimates, repeated variants, and local pairwise inconsistencies, so Seq2Graph serves as a data-integration pipeline rather than a graph-learning contribution. It uses graph edges for score propagation and consistency correction; recorded rounds, not graph paths, define chronological order. Seq2Graph combines within-round ordering with shared-variant anchors, producing sequence-defined activity scores instead of labels for planned designs or ancestry claims.

Using TadA-Bench, we find that representative biological language models struggle with future-round discovery. On the protein view, a random-split control reaches strong rank correlation, indicating that the label space is learnable under interpolation. Under future-round evaluation, however, DNA, RNA, and protein models degrade sharply, and finite-budget candidate selection remains weak. This gap shows that interpolation over known experimental distributions is not sufficient for prioritizing variants that emerge in later wet-lab rounds. The challenge exposed by TadA-Bench is therefore not merely an abstract distribution shift, but a concrete bottleneck for the sequence-ranking component of iterative protein-engineering workflows. To test whether this gap is an artifact of a minimal probing protocol, we include targeted adaptation and discovery-mode checks. Full fine-tuning and prompt tuning give the encoders more task-

---

†Corresponding authors. [1]Shanghai Jiao Tong University [2]Shanghai Innovation Institute. Correspondence to: Yuming Lu <luymin@sjtu.edu.cn>, Dequan Wang <dequanwang@sjtu.edu.cn>.

*Proceedings of the 43$^{rd}$ International Conference on Machine Learning*, Seoul, South Korea. PMLR 306, 2026. Copyright 2026 by the author(s).

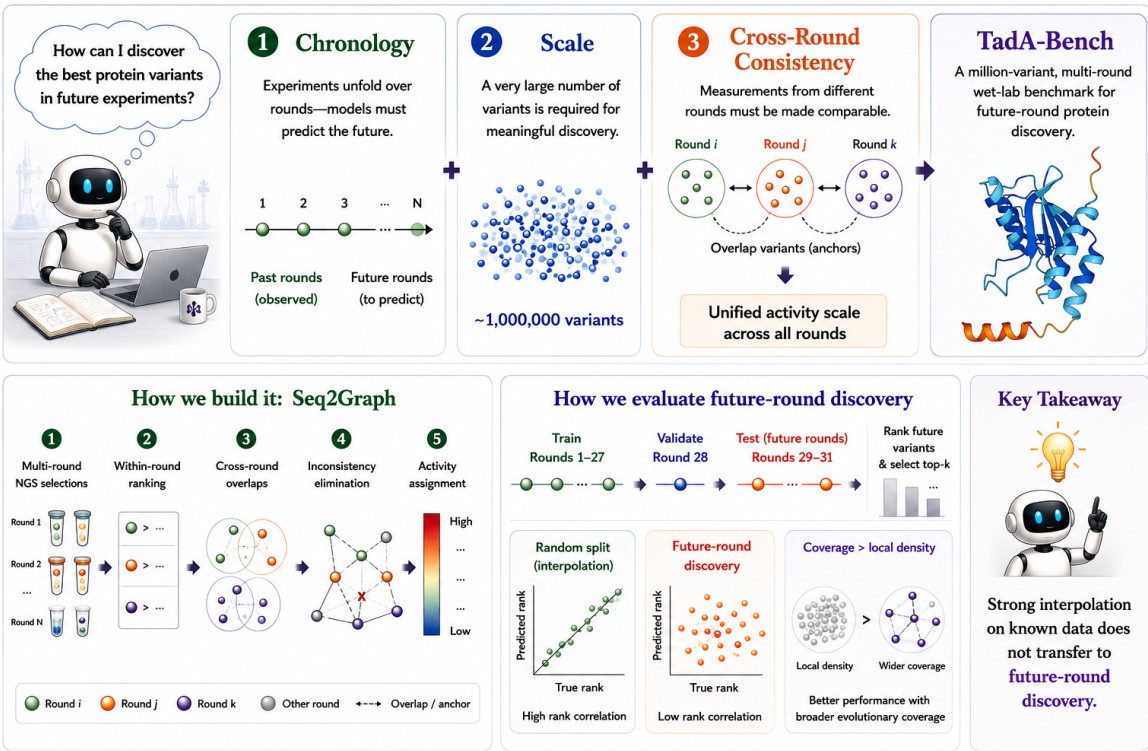

*Figure 1.* TadA-Bench overview. The benchmark frames protein engineering as fixed-data future-round discovery: models observe earlier wet-lab rounds and rank later-round variants. The design combines chronological replayability, million-scale variant coverage, and cross-round activity consistency. Seq2Graph builds a shared activity scale from multi-round NGS selections using within-round rankings, overlap anchors, inconsistency elimination, and activity assignment. Evaluation contrasts random-split interpolation with future-round ranking and finite-budget selection, showing that interpolation on known data does not ensure reliable future-round discovery.

specific flexibility, while finite-budget candidate prioritization asks how many true high-activity future variants a lab-facing agent would recover under a limited testing budget. The gap persists, indicating that the main difficulty is future-round prioritization from recorded evidence rather than the choice of a lightweight regression head.

Our controlled data analyses further suggest that future-round discovery benefits more from evolutionary coverage than from dense local sampling in matched-size analyses. We present this as an empirical observation on TadA-Bench rather than a universal law, but it reinforces the need to preserve campaign structure when building benchmarks for protein discovery. This distinction matters because matched-size controls separate broad campaign coverage from repeated sampling around already observed regions.

The benchmark remains fixed-data and reproducible: it isolates past-to-future ranking without evaluating planning policies, tool-use strategies, or autonomous wet-lab execution. This scope keeps TadA-Bench focused on the practical campaign decision of which variants should be prioritized next. It also gives model developers a stable test of whether representations use recorded experimental history before

proposal or acquisition layers are added. This keeps comparisons tied to a shared wet-lab history rather than rerun-specific experimental variation. The release follows the same principle: fixed splits, aligned sequence views, baseline protocols, and code let users compare models on one recorded campaign without rerunning wet-lab experiments.

**Contributions.**

- We introduce **TadA-Bench**, a 31-round, million-variant wet-lab benchmark for future-round replay with aligned DNA, RNA, and protein views.

- We develop **Seq2Graph**, a graph pipeline that unifies noisy multi-round enrichment into activity labels. Overlap anchors support label unification, not ancestry inference.

- We show that representative biological language models interpolate random splits yet fail on future-round ranking and finite-budget selection. Adaptation checks with full fine-tuning and prompt tuning do not close the gap.

- We find that evolutionary coverage is more informative than local data density on TadA-Bench. The result highlights coverage-aware benchmark design.

## 2. Related Work

### 2.1. Protein Activity Benchmark

While structural benchmarks in protein research have achieved notable success (Moult et al., 2020; Haas et al., 2018; Buttenschoen et al., 2024; Ye et al., 2025), functional benchmarks are still in nascent stages. These benchmarks are primarily categorized into two groups (West-Roberts et al., 2024): biophysical properties and deep mutational scanning (DMS) data. Benchmarks for biophysical properties (Bairoch, 2000; Xu et al., 2022; Zhou et al., 2019; Nikam et al., 2021; Rao et al., 2019; Vander Meersche et al., 2024) include enzymatic activity, fluorescence, thermodynamics, and solubility, but their broad focus limits their utility for precise future-round evaluation. DMS benchmarks (Fowler & Fields, 2014; Jiang et al., 2024; Gray et al., 2018) use large-scale mutagenesis and high-throughput sequencing to provide detailed views of local fitness landscapes. ProteinGym and FLIP aggregate diverse DMS datasets into broad fitness-prediction benchmarks (Notin et al., 2023; Dallago et al., 2021; Riesselman et al., 2018), and ProteinBench broadens evaluation across protein foundation-model tasks (Ye et al., 2025). TadA-Bench is complementary to these broad-coverage resources (Ding et al., 2024; Bozkurt et al., 2024; Notin et al., 2024): instead of maximizing the number of protein families or assay types, it follows one TadA engineering campaign in depth, preserving 31 wet-lab rounds, a fixed future-round split, and cross-round measurement consistency for evaluating whether models can rank variants from later experimental rounds in a single-campaign replay setting.

### 2.2. Base Editing Dataset

Current datasets focusing on deaminase enzyme optimization in base editing are relatively sparse and often fragmented. Most available resources (Dixit et al., 2024; Yan et al., 2020; Marquart et al., 2021; Sánchez-Rivera et al., 2022; Xiang et al., 2021; Leenay et al., 2019) emphasize the optimization of base editors through their interactions with CRISPR-associated proteins (Cas), single-guide RNAs (sgRNAs), or editing outcomes, rather than through systematic exploration of the deaminase variants themselves. These datasets are typically derived from narrow experimental conditions, thereby limiting their generalizability and scalability for machine-learning-based modeling and prediction. Several research groups have published improved or novel deaminase protein sequences through directed evolution, rational design, or machine-learning-guided design (Richter et al., 2020; Li et al., 2020; Tu et al., 2022; Perrotta et al., 2025; Cheng et al., 2024), but these studies usually use different wet-lab experimental conditions. Recent aggregation platforms such as CRISPRbase (Fan et al., 2023) aim to centralize base-editing datasets across publications and labs; while useful for accessibility, such aggregation can introduce batch effects due to diverse experimental protocols (Notin et al., 2023). To the best of our knowledge, no public TadA-focused benchmark provides a comparable combination of 31-round evolutionary depth, standardized label construction, million-scale sequence coverage, and a fixed future-round prediction protocol within one assay context and recorded experimental trajectory.

### 2.3. Biological Foundation Models

Biological foundation models provide the main model class evaluated by modern protein and genomics benchmarks. DNA and genome models such as Nucleotide Transformer and Evo 2 expand training scale, genomic coverage, and sequence context length (Dalla-Torre et al., 2025; Brixi et al., 2026); RNA models such as OmniGenome and Ri-NALMo incorporate RNA-specific sequence and structural signals (Yang et al., 2025; Penić et al., 2025); and protein language models have progressed from ESM2-scale representation learning to ESM Cambrian representations and ESM3-style generative modeling (Lin et al., 2023; ESM Team, 2024; Hayes et al., 2025). These models improve representation quality and broaden biological coverage, but TadA-Bench asks a narrower benchmark question: whether such representations support future-round ranking on a fixed wet-lab TadA engineering trajectory.

## 3. Benchmark Construction

This section turns the TadA campaign into a fixed-data wet-lab replay substrate for future-round discovery toward agentic protein engineering. The construction follows the three properties introduced above: chronological replayability defines the benchmark interface, exploration scale comes from the million-variant directed-evolution campaign, and cross-round label consistency is produced by Seq2Graph. The wet-lab protocol is described in Appendix A. Here we focus on the benchmark interface and the computational integration procedure summarized in Figure 2 for the main benchmark view used for evaluation.

### 3.1. Fixed-Data Future-Round Replay

Let $D_r$ denote the variants observed in experimental round $r$ of the TadA campaign. Each example consists of a biological sequence, a relative activity label constructed by Seq2Graph, and the recorded round metadata. A future-round replay instance is defined by a cutoff round $k$. A model observes variants and measurements from earlier rounds $D_{\leq k}$ and is evaluated by ranking variants that appear only in later rounds. This protocol asks whether past wet-lab evidence can prioritize future wet-lab outcomes within the same experimental campaign. It is a fixed candidate-ranking problem on recorded data; it does not simulate proposing

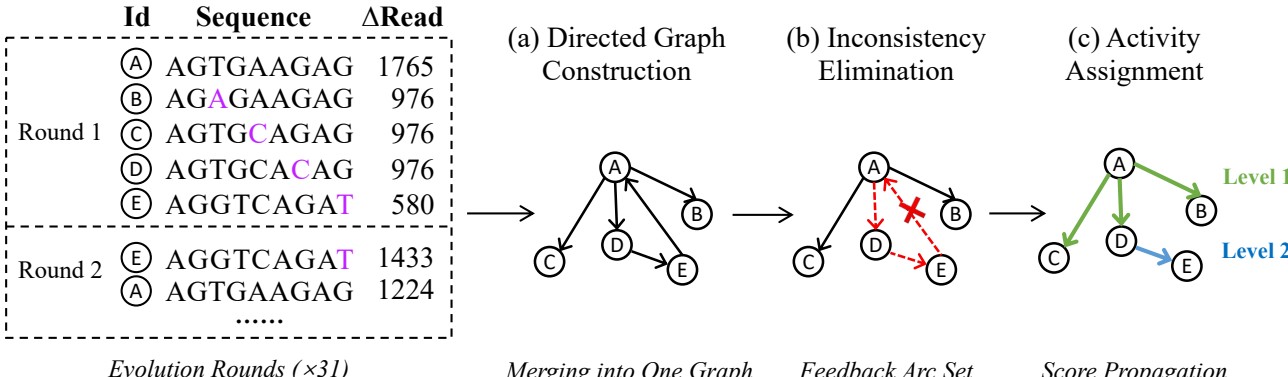

**Figure 2.** Seq2Graph constructs a unified activity scale from multi-round TadA sequencing data. (a) NGS enrichment within each PANCE round provides local rankings, and overlap variants connect rounds as anchors. (b) Weighted Feedback Arc Set correction removes contradictory cycles from noisy comparisons. (c) Scores are anchored at TadA8e and propagated as log-ratio evidence to assign sequence-defined activity labels for benchmark evaluation; the graph is not used for biological ancestry or experimental-lineage inference.

unseen variants, choosing experimental budgets, or executing wet-lab actions. In the main benchmark instance used throughout this paper, models train on rounds 1–27, validate on round 28, and test on rounds 29–31. The train, validation, and test sets are sequence-disjoint in the released views, so future-round evaluation cannot be solved by exact sequence memorization across released splits.

### 3.2. Dataset Scope and Sequence Views

The released TadA-Bench data are sequence-defined rather than design-intent-defined. Labels are assigned to variants observed by NGS after quality control, not only to variants listed in an intended mutation design. The libraries are generated by expert-designed degenerate synthesis and may contain multiple simultaneous mutations. Therefore, many variants are combinatorial mutants rather than single-edit descendants. Presence in the screened population does not imply that a sequence is "functional" under a binary threshold. Instead, each sequence receives a continuous relative activity value under the selection-coupled TadA assay. This value reflects the end-to-end cellular performance of the variant, including expression, folding or stability, and editing activity, rather than an isolated catalytic constant. The million-variant scale refers to NGS-observed DNA/RNA variants. After collapsing synonymous DNA sequences, the protein view contains 409,869 distinct amino-acid sequences. TadA-Bench provides aligned DNA, RNA, and protein views so that genomic, transcript, and protein language models can be evaluated on the same wet-lab campaign. The recorded round metadata are used to define the replay protocol; Seq2Graph does not infer biological ancestry or a lineage tree in the released benchmark.

### 3.3. Cross-Round Label Unification

Cross-round label consistency cannot be obtained by simply pooling normalized read counts from all experimental rounds. Even under a standardized wet-lab protocol, multi-round selection data contain batch effects, noisy enrichment estimates, repeated variants, and local inconsistencies. Seq2Graph therefore uses relative comparisons rather than absolute cross-round normalization. Within each round, variants are ranked by read-count enrichment. We build a directed graph $G = (V, E)$ where each unique DNA sequence is a node. For each round, we add edges only between adjacent variants in the sorted enrichment list, pointing from the higher-enrichment variant to the lower-enrichment variant. The edge weight is the local enrichment ratio. This local construction reduces sensitivity to absolute batch effects and keeps the graph sparse enough for million-scale integration. Exact sequence overlap across rounds connects the local ranking graphs. These overlaps act as anchors that allow relative activity information to propagate across the 31-round campaign. The experimental round is stored as node metadata; it is not inferred from graph distance.

### 3.4. Inconsistency Elimination

Aggregating noisy local rankings from many rounds can produce cycles, such as $v_i > v_j$, $v_j > v_k$, and $v_k > v_i$. Such cycles correspond to contradictory pairwise activity comparisons. To obtain a globally consistent ranking graph, we remove a low-confidence set of edges and transform the graph into a directed acyclic graph. We formulate this cleaning step as a weighted Feedback Arc Set problem:

$$\min_{F \subseteq E} \sum_{e \in F} w_e$$

$$\text{s.t.} \quad G' = (V, E \setminus F) \text{ is acyclic.}$$

(1)

Since exact Feedback Arc Set optimization is NP-hard, we use a greedy heuristic (Eades et al., 1993) within strongly connected components. This step should be viewed as practical consistency correction for benchmark construction, not as a new graph-theoretic primitive.

### 3.5. Activity Assignment

We anchor the activity scale by assigning TadA8e (Richter et al., 2020) a value of 1.0. Activities are propagated in the log domain because enrichment ratios are multiplicative. When a sequence is reachable through multiple paths, we use a fewest-edge path from the reference to reduce accumulated comparison noise. This path-selection step is used only for score propagation on the cleaned graph; it should not be interpreted as biological ancestry, evolutionary distance, or a lineage estimate. This process yields DNA-level activity labels. RNA sequences are obtained by replacing thymine with uracil. Protein-level labels are produced by mapping DNA variants to amino-acid sequences and averaging synonymous DNA-level activities. The synonymous aggregation is non-trivial because codon usage can affect the cellular assay, but it provides a stable protein-level target for protein language models in the protein view.

### 3.6. Construction Scope

Seq2Graph can be applied when multiple high-throughput screens provide partially overlapping sequence sets and local quantitative rankings. We include a Cas9 construction example in Appendix B.4 only to illustrate that the integration pipeline is not intrinsically TadA-specific. All benchmark claims and model-evaluation conclusions in this paper are restricted to TadA in this study.

## 4. Experiments

This section evaluates future-round discovery on TadA-Bench by treating language models as the ranking module a future agentic protein-engineering loop would consult before wet-lab follow-up. We compare future-round replay with a random-split control, test discovery-mode short-list prioritization under limited budgets, analyze density/diversity/round subsets, and validate the assay signal and Seq2Graph robustness. All experiments are fixed-data replays; they do not simulate proposal, planning, tool use, or autonomous wet-lab execution.

### 4.1. Experimental Setup

**Dataset.** We use the main fixed replay instance of TadA-Bench. Models train on rounds 1–27, validate on round 28, and test on rounds 29–31. The nucleic-acid views contain 729,302 training sequences, 148,014 validation sequences, and 149,884 test sequences. The protein view contains

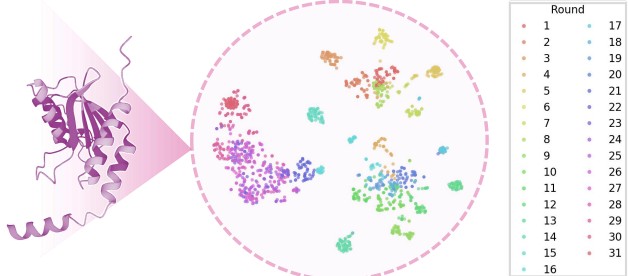

*Figure 3.* TadA structure and round-level variant landscape. *Left*: TadA structure predicted using ESMFold (Lin et al., 2023). *Right*: t-SNE visualization of variants sampled from the 31 experimental rounds, with point colors indicating experimental rounds. Later rounds need not move monotonically away from round 1; guided evolution creates local round clusters within the same TadA scaffold rather than a separate fold-level transfer problem.

256,429 training sequences, 45,208 validation sequences, and 108,232 test sequences. The train, validation, and test sets are sequence-disjoint in each released view. No additional wet-lab data are assumed during evaluation.

**Task and metrics.** The task is to predict relative activity scores for held-out variants and use these scores to rank future-round candidates. We evaluate ranking quality using Spearman's rank correlation $\rho$, Recall@10%, and nDCG@10% (Notin et al., 2023). Spearman's $\rho$ measures global monotonic ranking agreement. Recall@10% measures whether true top-decile variants are recovered in the predicted top decile. nDCG@10% measures ranking quality within the top-decile region used for candidate prioritization and finite-budget follow-up selection.

**Models.** We evaluate biological language models spanning DNA, RNA, and protein views. For DNA, we include Evo 2 (Brixi et al., 2026) and Nucleotide Transformer (Dalla-Torre et al., 2025) models. For RNA, we evaluate OmniGenome models (Yang et al., 2025). For proteins, we evaluate ESM2 (Lin et al., 2023), ProtTrans (Elnaggar et al., 2021), and ESMC (ESM Team, 2024). The primary protocol freezes each encoder and trains the same downstream regression head. Because the sequence views differ across modalities, direct cross-modality ranking of model families is not the focus. Additional split audits and implementation details are provided in Appendix C.

### 4.2. Interpolation Does Not Imply Future-Round Discovery

The central question is whether models that interpolate well over observed variants can also rank variants from later wet-lab rounds. The round-wise visualization in Figure 3 shows that the 31-round campaign contains structured clusters across experimental rounds within the same TadA scaf-

*Table 1.* Future-round performance on the nucleic-acid views of TadA-Bench. Models train on rounds 1–27, validate on round 28 for learning-rate selection, and test on rounds 29–31. **Bold** marks the best value for each metric. DNA models use the DNA view; OmniGenome (OG) uses the T-to-U RNA view for the same split protocol.

| Model | Validation | | | Test | | |
|---|---|---|---|---|---|---|
| | Spearman | Recall@10% | nDCG@10% | Spearman | Recall@10% | nDCG@10% |
| Evo2-7B | 0.0490 | 0.1097 | 0.2604 | **0.0707** | 0.1005 | 0.3236 |
| Evo2-40B | **0.0980** | **0.1157** | **0.2702** | 0.0675 | 0.1003 | **0.3244** |
| NT-50M | 0.0401 | 0.0959 | 0.2464 | 0.0166 | 0.0950 | 0.3109 |
| NT-100M | 0.0520 | 0.0982 | 0.2485 | 0.0045 | 0.0870 | 0.3048 |
| NT-250M | 0.0470 | 0.0858 | 0.2137 | 0.0006 | 0.0971 | 0.3085 |
| NT-500M | 0.0361 | 0.0985 | 0.2225 | 0.0189 | 0.1005 | 0.3079 |
| OG-46M | 0.0555 | 0.0911 | 0.2192 | 0.0079 | **0.1063** | 0.3158 |
| OG-418M | 0.0078 | 0.0949 | 0.2391 | 0.0048 | 0.0859 | 0.3042 |

*Table 2.* Future-round performance on the protein view of TadA-Bench. Models train on rounds 1–27, select the learning rate on round 28, and test on rounds 29–31. **Bold** numbers mark the best value for each metric.

| Model | Validation | | | Test | | |
|---|---|---|---|---|---|---|
| | Spearman | Recall@10% | nDCG@10% | Spearman | Recall@10% | nDCG@10% |
| ESM2-35M | **0.1591** | 0.1449 | 0.6533 | 0.0300 | **0.1379** | **0.3281** |
| ESM2-150M | 0.1458 | 0.1420 | 0.6569 | 0.0416 | 0.1230 | 0.3068 |
| ESM2-650M | 0.1423 | **0.1473** | 0.6530 | 0.0479 | 0.1120 | 0.2791 |
| Prot-BERT | 0.1280 | 0.1128 | 0.6534 | 0.0214 | 0.1162 | 0.2980 |
| Prot-XLNET | 0.1570 | 0.1261 | **0.6589** | 0.0342 | 0.1175 | 0.2895 |
| ESMC-300M | 0.1498 | 0.1199 | 0.6495 | 0.0355 | 0.1151 | 0.2867 |
| ESMC-600M | 0.1452 | 0.1206 | 0.6397 | **0.0509** | 0.1180 | 0.2860 |

fold. This visualization is qualitative; the quantitative test is the fixed future-round replay split.

Under the future-round replay split, biological language models exhibit consistently weak ranking performance. As shown in Tables 1 and 2, frozen-encoder probing Spearman correlations are only around $\rho \approx 0.1$ or below across DNA, RNA, and protein views. This indicates weak past-to-future extrapolation by current representations. The pattern is consistent across DNA, RNA, and protein views, so the gap is not tied to one sequence encoding. Because the released splits are sequence-disjoint, the low correlations reflect weak ranking on recorded future evidence rather than exact-sequence leakage or split contamination.

The random-split control gives a different picture. As shown in Table 4, protein language models reach Spearman correlations near $\rho \approx 0.8$ under an 8:1:1 random split. The same label space is therefore learnable under interpolation. The sharp contrast in Figure 4 shows that random-split success does not imply future-round discovery. This contrast separates label quality from past-to-future generalization: the labels are learnable under interpolation, but representations that organize known variants well still fail when evidence is restricted to earlier experimental rounds.

**Targeted adaptation checks.** The headline comparison uses frozen encoders to evaluate the representations under a

*Table 3.* Targeted adaptation checks on the future-round split. Each row reports an upper-bound diagnostic over the checked hyperparameters, not a leaderboard result; full results are provided in the appendix for reproducibility.

| Adaptation | Checked setting | Best observed test $\rho$ |
|---|---|---|
| Full fine-tuning | ESM2 family | 0.055 |
| Full fine-tuning | NT family | 0.063 |
| Prompt tuning | NT-50M | 0.075 |
| Prompt tuning | ESMC-300M | 0.027 |

shared downstream head. To test whether the future-round gap is merely an artifact of this lightweight protocol, we also evaluate representative full fine-tuning and prompt-tuning settings. As summarized in Table 3, these targeted adaptation checks remain in the same low-Spearman regime as frozen-encoder probing. Full hyperparameter tables are reported in Tables 7 and 8. These checks are reported as diagnostics rather than model-selection results; even favorable adaptation settings do not remove the observed future-round gap, reinforcing that the benchmark difficulty is not an artifact of the frozen-encoder protocol.

### 4.3. Discovery-Mode Candidate Prioritization

Future-round ranking matters in discovery mode because only a short test list can be sent to wet-lab follow-up. We

*Table 4.* Random-split interpolation performance on the protein view of TadA-Bench. Data are split 8:1:1 at random, and validation performance selects the learning rate. **Bold** numbers mark the best value for each metric.

| Model | Validation | | | Test | | |
|---|---|---|---|---|---|---|
| | Spearman | Recall@10% | nDCG@10% | Spearman | Recall@10% | nDCG@10% |
| ESM2-35M | 0.8032 | 0.1830 | 0.4824 | 0.8014 | 0.1617 | 0.4814 |
| ESM2-150M | 0.7386 | 0.2290 | 0.4364 | 0.7371 | 0.2324 | 0.4437 |
| ESM2-650M | 0.5360 | 0.1793 | 0.4740 | 0.5348 | 0.1710 | 0.4779 |
| Prot-BERT | 0.7910 | 0.2230 | 0.4879 | 0.7883 | 0.2262 | 0.4918 |
| Prot-XLNET | 0.8054 | 0.2264 | 0.4912 | 0.8030 | 0.2193 | 0.4965 |
| ESMC-300M | 0.8102 | 0.2439 | 0.4959 | 0.8067 | **0.2363** | **0.4995** |
| ESMC-600M | **0.8127** | **0.2446** | **0.5006** | **0.8079** | 0.2317 | 0.4949 |

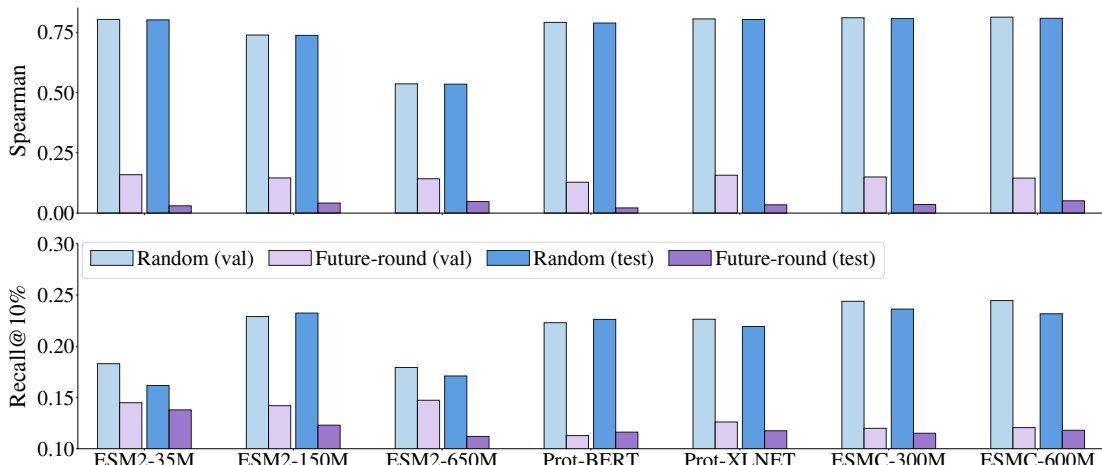

*Figure 4.* Random-split interpolation versus fixed future-round evaluation on the protein view of TadA-Bench. The top and bottom panels report Spearman's $\rho$ and Recall@10%, respectively. "Random" denotes the 8:1:1 random split, while "Future-round" denotes the fixed replay split that trains on earlier rounds and evaluates later rounds. Shallow bars denote validation performance and deeper bars denote test performance for the same models.

therefore treat each model as an offline ranking module and ask whether its top-ranked later-round variants are enriched for true high-activity variants under small budgets. It should not be interpreted as a deployable acquisition policy, because the candidate set and wet-lab outcomes are fixed in advance. We evaluate budgets of 96 candidates, corresponding to a standard 96-well plate, and 384 candidates. For representative protein baselines, neither budget selects any true top-1% future-round variants. Their top-5% hit rates are also close to random: 3.1% and 4.9% at $K = 96$, and 4.3% and 4.2% at $K = 384$, for ESM2-35M and ESMC-300M, respectively, compared with the 5% random expectation. Thus, the future-round gap is not only visible in aggregate rank correlation. It also translates into weak discovery-mode prioritization of later-round variants.

### 4.4. Coverage Helps More Than Local Density

Having established the future-round ranking gap, we next ask what type of training evidence helps close it. We do not claim a universal causal law that diversity or round depth always dominates volume. Instead, we test an empirical

hypothesis on this fixed benchmark: subsets that cover more functionally diverse or later evolutionary regions should be more informative than equally sized dense samples from already observed regions. These subset constructions are validation-aware diagnostics of training-set composition, not deployable acquisition policies. We keep validation and test sets fixed and vary the training subset with three matched-split strategies.

**Density:** A random subsample of the full training set. This simulates simply collecting more data from the same evolutionary rounds without changing coverage.

**Diversity:** Sequences selected from all training rounds to maximize similarity with the validation set. This prioritizes sequence regions closer to the target split under the chosen similarity measure.

**Round:** Entire experimental rounds selected based on their aggregate similarity to the validation set. This preserves intra-round correlations while moving closer to the target task under the replay protocol.

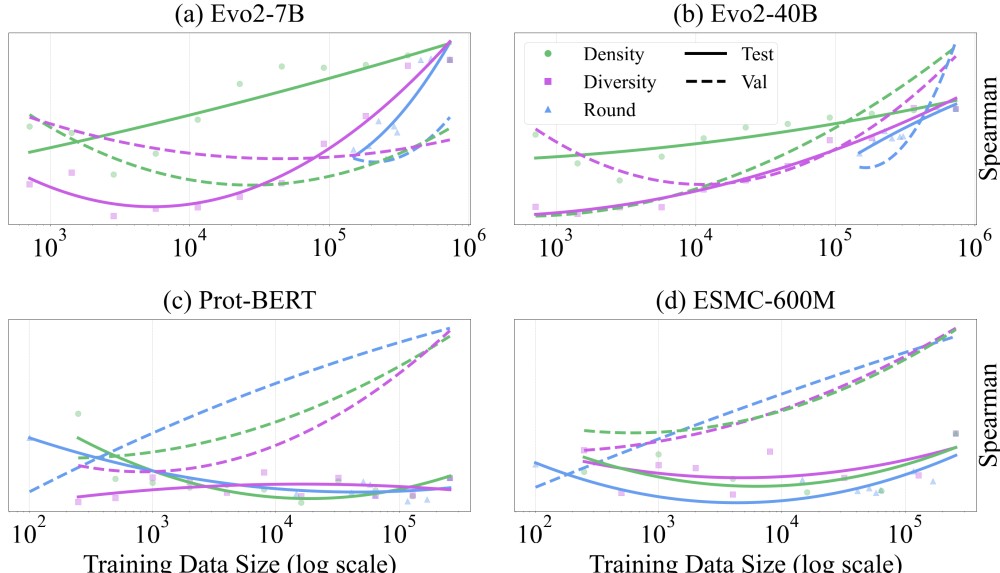

*Figure 5.* Performance under training-set construction strategies. Density randomly subsamples training variants, diversity selects variants similar to the validation set, and round selects whole experimental rounds by aggregate similarity. The x-axis is training-set size on a log scale and the y-axis denotes Spearman's $\rho$; because round-based selection uses whole rounds, its smallest bundle can exceed $10^5$ sequences while preserving round structure for comparison across strategies.

These diagnostics use sequence similarity only; they do not use validation or test activity labels for model training.

As shown in Figure 5, performance increases with training-set size, but the scaling behavior depends strongly on subset construction. Across the tested models, the *diversity* and *round*-based strategies consistently outperform the naive *density*-based strategy. In several cases, a smaller diversity-focused subset outperforms a much larger randomly sampled subset. This suggests that, on TadA-Bench, future-round discovery benefits more from covering relevant evolutionary or functional regions than from increasing local sample density. From the benchmark-design perspective, the result supports preserving campaign structure rather than treating all additional variants as equally informative.

### 4.5. Label Consistency and Construction Robustness

Because TadA-Bench relies on cross-round label consistency, we validate both the biological assay signal and the computational robustness of Seq2Graph. The goal is to confirm that the future-round gap is not an artifact of meaningless labels or an unstable construction pipeline.

For biological validation, we performed orthogonal low-throughput measurements using a GFP reporter assay. We selected key variants spanning a range of TadA-Bench activity scores and independently measured their activities. The rank ordering from the GFP assay strongly agrees with the ordering derived from high-throughput NGS enrichment, with Spearman's $\rho > 0.99$. This supports the biological

meaningfulness of the high-throughput activity labels.

For computational validation, we conduct two bootstrapping analyses. First, we randomly subsample 50% of the experimental rounds and rerun the entire construction pipeline. The resulting activity scores for shared sequences have Spearman's $\rho = 0.90$ ($p < 10^{-5}$). Second, we subsample 50% of sequences within each round before rerunning the pipeline, obtaining Spearman's $\rho = 0.95$ ($p < 10^{-5}$). These results indicate that Seq2Graph is robust to variations in round coverage and sequencing depth. Together, the biological and computational validations support the reliability of TadA-Bench as a future-round replay benchmark.

## 5. Conclusion

We introduced TadA-Bench as a million-variant wet-lab replay benchmark for future-round discovery toward agentic protein engineering, preserving recorded chronology, broad variant coverage, and Seq2Graph-based label consistency while exposing a future-round ranking gap hidden by random splits. The benchmark isolates the ranking module that future agentic protein-engineering systems will need before proposal, acquisition, tool use, wet-lab coordination, or autonomous execution, providing a controlled offline replay test before costlier closed-loop campaigns.

**Limitations.** TadA-Bench remains a fixed-candidate TadA assay with cellular activity labels; future extensions can add proteins and assays while preserving future-round replay toward agentic protein engineering.

## Acknowledgements

This research is supported by the Key R&D Program of Shandong Province, China (2024CXGC010213, 2023CXGC010112). We express our gratitude to the funding agency for their support.

## Impact Statement

TadA-Bench provides a fixed-data future-round replay benchmark toward agentic protein engineering, with the intended benefit of making model comparisons more reproducible before costly wet-lab follow-up. By releasing sequence-defined data, fixed splits, labels, baseline protocols, and evaluation code, the benchmark helps study chronological generalization and candidate ranking without requiring new experiments or autonomous wet-lab execution. The data come from a controlled TadA engineering campaign and do not include complete actionable protocols for harmful biological deployment, but protein-engineering models can still have dual-use implications; users should follow institutional biosafety, biosecurity, and ethical guidelines when applying models trained or evaluated with TadA-Bench in downstream protein-engineering studies.

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

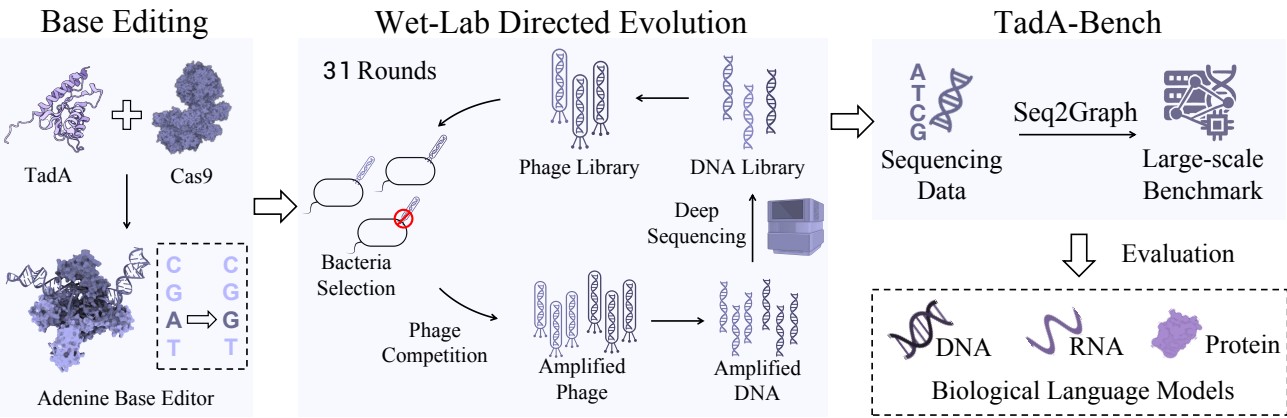

*Figure 6.* Wet-lab and benchmark-construction workflow for TadA-Bench. *Left*: TadA is the deaminase component used in base editing. *Center*: 31 rounds of PANCE selection generate large-scale sequencing data. *Right*: Seq2Graph converts the campaign into benchmark labels for evaluation with biological language models in the released benchmark.

# Appendix

This appendix provides supporting details for the wet-lab campaign, benchmark construction, and model evaluation. The authors used large language models (LLMs) for language editing, formatting assistance, and visualization suggestions during manuscript preparation; all scientific claims, experimental design, analyses, and final manuscript content were developed, verified, and approved by the authors. Appendix A describes the wet-lab assay and data-collection workflow that produced the 31-round TadA campaign. Appendix B provides implementation details for Seq2Graph, the label-unification pipeline used to construct consistent cross-round activity scores. Appendix C reports experimental details, split audits, and additional adaptation checks.

## A. Wet-Lab Assay and Data Collection

We provide a visual summary of our entire pipeline, from the biological experiments to the final benchmark construction, illustrated in Figure 6 before the detailed sections that follow in this appendix.

### A.1. Biological and Experimental Preliminaries

**Adenine Base Editing and TadA:** Base editing enables precise single-nucleotide conversions without the double-strand breaks (DSBs) associated with traditional CRISPR methods, offering improved safety and precision (Cox et al., 2015; Hilton & Gersbach, 2015; Komor et al., 2016; Gaudelli et al., 2017). Adenine base editors (ABEs), the focus of this work, convert A•T to G•C base pairs. This is achieved using an engineered tRNA-specific adenosine deaminase (TadA) enzyme, which catalyzes the deamination of adenosine (A) to inosine (I), an intermediate interpreted as guanosine (G) during DNA replication. Our work centers on TadA8e (Richter et al., 2020), a high-activity variant comprising 167 amino acids. Our benchmark, TadA-Bench, is constructed from libraries of TadA8e variants generated through extensive mutagenesis.

**PANCE for High-Throughput Activity Annotation:** We employ Phage-Assisted Non-Continuous Evolution (PANCE) (Miller et al., 2020; Zhang et al., 2024), a directed evolution method, to screen TadA variants for their activity at scale. In this system, a library of TadA variants is encoded within a bacteriophage population. Phages carrying variants with higher enzymatic activity replicate more efficiently, leading to their enrichment. The relative abundance of each variant is then quantified via Next-Generation Sequencing (NGS), where higher read counts or enrichment provide local within-round evidence for activity. These local signals are later unified by Seq2Graph rather than pooled directly across rounds. Our TadA-Bench dataset comprises data from 31 recorded PANCE rounds/screens within a single TadA engineering campaign, each screening a unique library of tens of thousands of TadA variants.

**Degenerate Sequences for Efficient Library Synthesis:** Synthesizing large and diverse variant libraries by creating each DNA sequence individually is prohibitively expensive. To overcome this, we utilize degenerate sequence synthesis (Li et al., 2022). A degenerate sequence is a single oligonucleotide template that contains ambiguous nucleotide codes (*e.g.*,

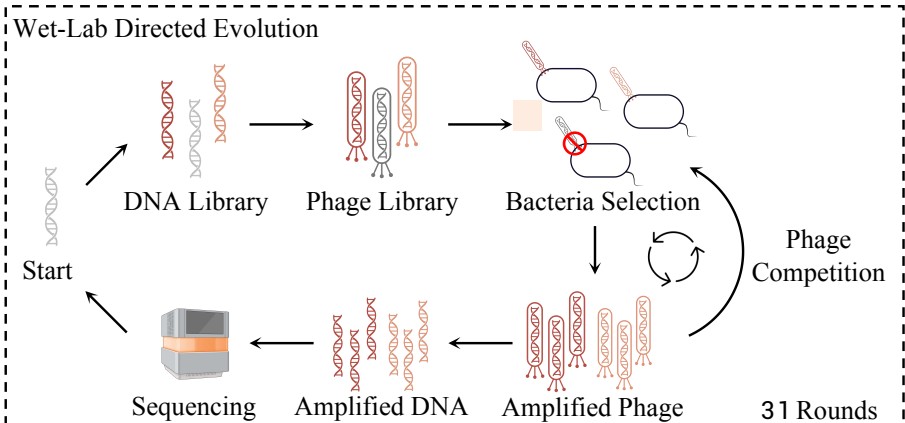

*Figure 7.* PANCE workflow for TadA activity annotation. Degenerate libraries create TadA variants, and phage-assisted selection converts cellular activity into enrichment. More active variants activate *gIII* expression more strongly, propagate more efficiently, and become enriched across PANCE cycles before sequencing for downstream activity annotation.

'N' representing A, C, G, or T) at specified positions. This technique enables the cost-effective generation of a vast library containing thousands of unique variants from a single synthesis reaction, which was essential for constructing the diverse libraries screened in our PANCE experiments at million-variant scale.

## A.2. Experimental Workflow for Large-Scale Activity Annotation

The goal of the wet-lab workflow was not only to enrich active TadA variants, but also to produce a recorded experimental campaign that can be converted into a fixed future-round replay benchmark. The workflow therefore needed to preserve experimental-round metadata, explore a large variant space, and maintain sufficient measurement consistency for cross-round label unification in the benchmark rather than only within individual rounds.

We used a multi-stage PANCE workflow that integrates expert-guided library design, high-throughput selection, deep sequencing, and orthogonal validation. The same campaign produced high-activity TadA variants, indicating that the selection system was biologically active; the benchmark labels are further supported by the reporter validation below and by the label-consistency analysis in Section 4.5 of the main text. Figure 7 summarizes the PANCE selection workflow.

### A.2.1. EXPERT-GUIDED LIBRARY DESIGN

A key departure from standard continuous evolution systems that rely on random mutagenesis is our use of expert-designed libraries. To ensure the utility and generality of the future-round data, we adopted a well-established pipeline common in industrial protein engineering. Each of the 31 experimental rounds began with a unique, rationally designed library of TadA variants. Protein engineers, leveraging structural information and domain expertise, identified promising regions for mutation and designed targeted libraries using degenerate sequence synthesis (*e.g.*, using NNK codons). For benchmark construction, this approach serves two critical functions that connect library design to replay evaluation:

**Reflects applied engineering practices.** Expert-designed degenerate synthesis introduces a knowledge-driven bias that focuses exploration on promising regions of the sequence space. This strategy mirrors common industrial protein-engineering practice, which often favors targeted mutagenesis over uniformly random mutagenesis. It also permits multiple simultaneous mutations: a single degenerate template can generate combinatorial variants rather than single-edit descendants. Because the realized library may include sequences outside the intended design, TadA-Bench is sequence-defined: labels are assigned to NGS-observed TadA sequences after quality control, not to design intents or planned mutation templates. The chronological order used by the benchmark comes from recorded experimental-round metadata, not from an assumed single-edit lineage.

**Establishes a controlled measurement baseline.** Using a shared degenerate-synthesis and selection workflow helps reduce large uncontrolled differences in initial library construction and simplifies downstream enrichment-based activity estimation. This control is important because TadA-Bench uses relative activity values from a multi-round selection campaign rather than isolated low-throughput kinetic measurements.

### A.2.2. PANCE SELECTION

PANCE was used as the high-throughput selection engine for assigning relative activity signals to TadA variants. The core principle is to couple the function of the target protein to bacteriophage replication. In our system, the M13 phage lacks the essential gene *gIII*, which encodes the pIII protein required for virion release. The expression of *gIII* is controlled by a reporter activated by TadA activity. Consequently, phages encoding more active TadA variants replicate more efficiently and become enriched in the population over selection cycles of the assay.

To ensure a robust selection process, each of the 31 rounds involved five cycles of serial dilution and competitive regrowth. This iterative enrichment minimizes stochastic effects and ensures the final phage population's composition reflects the relative activity of the encoded variants. Phages encoding high-activity proteins replicate more efficiently, ultimately dominating the population, while those encoding low-activity variants are progressively eliminated.

### A.2.3. ACTIVITY QUANTIFICATION AND ROBUSTNESS

We quantified the activity of each TadA variant based on its read count enrichment, as determined by deep Next-Generation Sequencing (NGS) of the enriched phage population. To guarantee precision and minimize sampling noise, each round was sequenced at an exceptional depth of over 100G bases, ensuring accurate quantification of variant frequencies.

**NGS error control.** We reduce technical sequencing artifacts by standard read-processing and read-support filtering before label construction. The downstream Seq2Graph integration further relies on local within-round comparisons and repeated cross-round sequence overlap rather than fragile absolute counts. The label-consistency analyses in Section 4.5 show that the resulting labels are stable under substantial removal of rounds or reads.

**Activity interpretation.** The measured value is an end-to-end cellular functional activity score. A low score may result from weak catalysis, poor expression, misfolding, instability, or other bottlenecks in the selection system. These factors are not treated as confounders for this benchmark; they are part of the functional phenotype that a protein-engineering model must rank under the benchmark protocol for future-round evaluation.

### A.2.4. ORTHOGONAL VALIDATION OF ACTIVITY LABELS

To address whether our high-throughput activity labels are biologically meaningful, we performed orthogonal validation using a low-throughput cellular reporter assay. We engineered a Green Fluorescent Protein (GFP) reporter system in which a mutated, non-fluorescent GFP gene could be repaired by the editing activity of a TadA variant, leading to a measurable fluorescent signal for independent validation of activity in the reporter assay.

We selected several key variants spanning a range of activity scores from the PANCE screens and independently measured their activity using this GFP assay. The results support ranking consistency: the rank ordering of variant activities determined by the GFP assay showed high agreement (*e.g.*, Spearman's rank correlation $\rho > 0.99$) with the rank ordering derived from our high-throughput NGS read-count enrichment. This confirmation supports the biological relevance of the cellular activity labels used in TadA-Bench for ranking evaluation in the benchmark.

## B. Seq2Graph Construction Details

This section provides implementation details for Seq2Graph, complementing the overview in Section 3. Seq2Graph is used to integrate noisy multi-round enrichment measurements into comparable activity labels. It should be interpreted as a benchmark-construction pipeline for cross-round label unification, not as a graph-learning model or a method for inferring biological ancestry or experimental lineage in the wet-lab campaign.

### B.1. Directed Graph Construction

In our directed evolution experiments, the activity of a TadA variant is proportional to its replication rate, which is measured by read counts from Next-Generation Sequencing (NGS)[1]. A common practice is to normalize these counts to estimate variant activities (Wagner et al., 2012; Love et al., 2014; Robinson & Oshlack, 2010). However, while our 31 experimental

---

[1]Next-Generation Sequencing (NGS) is a technology platform that enables high-throughput sequencing. In this paper, we use NGS and deep sequencing interchangeably in the wet-lab workflow description.

rounds followed a standardized protocol, inherent biological and technical stochasticity introduces batch effects, making it impossible to aggregate normalized counts from different rounds into a single, consistent dataset.

To solve such limitations, we instantiate a directed-graph representation of relative variant activity, rather than relying on absolute activity after cross-round normalization, as shown in Figure 2 (a). In the directed graph, $G = (V, E)$, the DNA sequence of each variant obtained from NGS is taken as a node $v_i$. For the list of growth multiples for read counts, the number of edges in $G$ can be up to $|V|^2$. For million-scale construction, we require a sparse graph that remains weakly connected so that relative activity information can propagate across observed variants.

Specifically, we sort the list of growth multiples for read counts and only add edges for nodes with adjacent count values along the list. To further manage computational complexity, we cap the number of edges generated per experiment at 100,000. Each edge points from nodes with higher activity to those with lower activity, which means the edge weight is always greater than 1. The weight of edge $e_{i \to j}$ represents the relative activity of $v_i$ over $v_j$, $w_{ij} = \frac{C(v_i)}{C(v_j)}$, where $C(\cdot)$ denotes the read-count growth multiple used for within-round ranking evidence.

Since the directed evolution process iteratively enriches for high-performing variants, some sequences are present across multiple experimental rounds, acting as bridging nodes within the dataset. Consequently, the graph composed of all 31 experimental datasets is weakly connected, ensuring a relative activity path exists between nearly any two variants. We also store the experimental round as a node attribute; on average, each node appears in 1.58 rounds.

### B.2. Inconsistency Elimination

Aggregating pairwise comparisons from 31 rounds inevitably introduces conflicting relationships (*e.g.*, $v_i > v_j$, $v_j > v_k$, but $v_k > v_i$). These conflicts manifest as cycles, or strongly connected components (SCCs), in the graph, as shown in Figure 2 (b). To ensure a globally consistent activity ranking, these cycles must be eliminated.

We assume that edges with higher weights, corresponding to larger read count ratios, are more reliable as they are less susceptible to measurement noise. We use the weighted Feedback Arc Set (FAS) formulation as a consistency-cleaning objective: remove a low-confidence set of edges whose removal makes the graph acyclic, as shown in Equation (2).

$$\min_{F \subseteq E} \quad \sum_{e \in F} w_e$$
$$\text{s.t.} \quad G' = (V, E \setminus F) \text{ is acyclic} \tag{2}$$

As the exact FAS problem is NP-hard and our graph is large, we employ a scalable greedy approximation (Eades et al., 1993) within each SCC. The result is a directed acyclic graph (DAG) containing the original set of nodes but with fewer edges.

### B.3. Activity Assignment

With the DAG established, we assign a consistent, relative activity score to each node. We anchor the scale by setting the activity of the reference sequence (TadA8e (Richter et al., 2020)) to 1.0. Since phage replication exhibits exponential growth, we propagate activity scores in a log-additive fashion along the graph edges.

For each reachable node, we use a deterministic fewest-edge path from the reference sequence for score propagation on the cleaned graph. This path-selection step is not used to infer biological ancestry, experimental lineage, or evolutionary distance. Paths with fewer pairwise comparisons are less likely to accumulate experimental noise. The graph is treated as undirected only for path selection; score propagation still applies each edge's log-ratio with the appropriate direction. This process yields the DNA version of Seq2Graph, which contains 1,027,200 DNA sequences with their assigned activity labels.

Finally, to create a protein-level dataset, we map DNA-level activities to their corresponding protein sequences. Since multiple DNA variants can encode the same protein, we compute the final activity for each protein by averaging the activities of all its corresponding DNA sequences. This averaging is non-trivial, as synonymous DNA variants can exhibit different activities due to effects like codon usage bias. This final step produces our protein dataset, comprising 409,869 annotated protein sequences for protein-level evaluation in the released benchmark.

*Table 5.* Statistics for the auxiliary Cas9 construction example created by Seq2Graph. The 8:1:1 split summarizes the pipeline output for inspection only; Cas9 is not used as a benchmark task in this paper or in the reported model comparisons.

|         | Train  | Val   | Test  | Total  |
|---------|--------|-------|-------|--------|
| DNA     | 255918 | 31989 | 31991 | 319898 |
| RNA     | 255918 | 31989 | 31991 | 319898 |
| Protein | 131735 | 16466 | 16468 | 164669 |

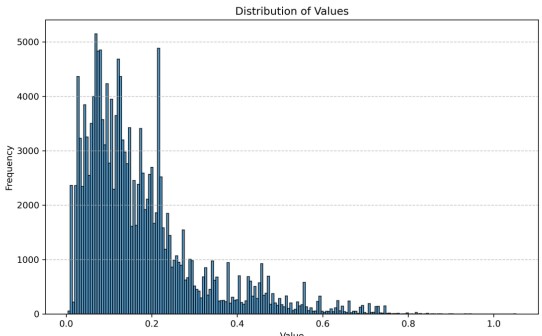 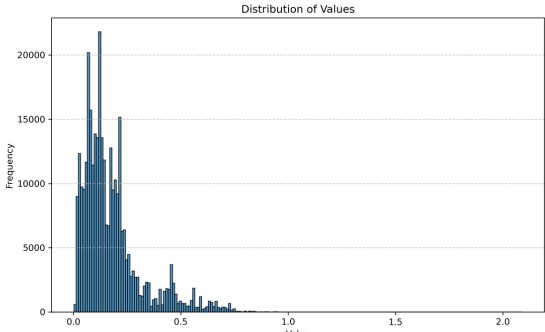

*Figure 8.* Activity-score distributions for the auxiliary Cas9 construction example created by Seq2Graph. The *left* panel shows protein-level scores, while the *right* panel shows DNA/RNA-level scores used for construction-scope inspection.

## B.4. Auxiliary Construction Scope Check on Cas9

When overlapping high-throughput screens provide compatible local rankings, the same integration procedure can be applied beyond TadA. To illustrate this construction scope, we apply Seq2Graph to a Cas9 screen as an auxiliary example, as shown in Table 5 and Figure 8. We do not treat Cas9 as a second benchmark in this paper, and we do not draw model-performance conclusions from it; all empirical benchmark claims are restricted to TadA.

## C. Experimental Details of TadA-Bench

### C.1. Dataset, Task, and Evaluation

This section expands the experimental protocol summarized in Section 4.1. TadA-Bench evaluates a fixed-data future-round replay problem: models use variants and activity labels from earlier recorded rounds to rank recorded variants from later rounds. The evaluation is offline and does not assume additional wet-lab data, variant proposal, acquisition policies, tool use, or autonomous execution during benchmark evaluation reported here.

**Dataset and Task.** TadA-Bench is derived from Next-Generation Sequencing (NGS) data of evolved TadA variants, available as both nucleic acid (DNA/RNA) and protein sequences (see Section 3.5 and Appendix B.3). Our primary task is the fixed future-round replay problem: models use earlier rounds to predict relative activities for recorded variants from later rounds and rank those candidates. Success is measured not by precise regression of activity values, but by the ability to correctly rank top-performing sequences. This is because experimental validation capacity is limited, making the accurate identification of a small set of promising candidates paramount.

**Data Splits.** We employ two distinct data-splitting strategies to evaluate model performance under different conditions.

- **Fixed Future-Round Split:** To instantiate the recorded-data replay protocol, we split the data by recorded round: rounds 1–27 for training, round 28 for validation, and rounds 29–31 for testing. The nucleic acid dataset contains 729,302 training, 148,014 validation, and 149,884 test sequences. The protein dataset comprises 256,429 training, 45,208 validation, and 108,232 test sequences for the released protein view of the benchmark.

- **Random Split:** To establish an idealized interpolation baseline, we randomly partition the entire dataset into training

*Table 6.* Audit of the fixed future-round split. Exact sequence overlap is zero across released DNA, RNA, and protein splits. Nearest-train diagnostics report normalized protein Hamming distance for 500 sampled validation or test queries against 5,000 sampled training sequences; the means correspond to about 2.5 and 5.3 amino-acid differences in 167-aa TadA.

| Audit | View | Train–val | Train–test | Val–test |
|---|---|---|---|---|
| | DNA | 0 | 0 | 0 |
| Exact split overlap | RNA | 0 | 0 | 0 |
| | Protein | 0 | 0 | 0 |

| Audit | Split | Mean | Median | P90 |
|---|---|---|---|---|
| Protein nearest-train normalized Hamming | Validation | 0.0148 | 0.0120 | 0.0180 |
| | Test | 0.0318 | 0.0299 | 0.0419 |

(80%), validation (10%), and test (10%) sets for interpolation analysis.

**Evaluation Metrics.** Following established practices in protein fitness prediction (Notin et al., 2023), we evaluate models using three key metrics. **Spearman's rank correlation coefficient ($\rho$)** measures the model's ability to capture the overall monotonic relationship between predicted and true activity scores. To assess performance on identifying top candidates, we use **Recall@10%**, the fraction of true top-10% variants identified in the predicted top 10%, and **normalized Discounted Cumulative Gain at 10% (nDCG@10%)**, which further evaluates the ranking quality within this top decile.

### C.2. Future-Round Split Audit

The future-round split is chronological and sequence-disjoint, but it should not be interpreted as a fold-level structural transfer task. Later rounds can remain close to earlier TadA variants while still representing future experimental evidence unavailable during training. We therefore audit the split using exact sequence overlap and sampled nearest-train protein Hamming diagnostics, reported in Table 6. These nearest-train diagnostics complement the t-SNE visualization in Figure 3: the split is shifted forward in time, but the shift remains within the same TadA scaffold and should be interpreted as library branches or mutational subspaces rather than a wholesale fold-level structural shift.

### C.3. Models and Implementation Details

**Pre-trained Models.** We evaluate a diverse suite of pre-trained biological language models (BLMs). For nucleic acids, we test models from the Evo 2 (Brixi et al., 2026) family (Evo2-7B, Evo2-40B) and NucleotideTransformer (Dalla-Torre et al., 2025) family (NT-50M, NT-100M, NT-250M, NT-500M). For proteins, we include models from the ESM2 (Lin et al., 2023), ProtTrans (Elnaggar et al., 2021), and ESMC (ESM Team, 2024) families, with concrete model rows reported in the main tables. When using RNA-specific models, DNA sequences were processed by mapping Thymine (T) to Uracil (U).

**Frozen-Encoder Probe Implementation.** For our primary evaluation, we perform frozen-encoder probing on features extracted from pre-trained models. For most models, we use the final hidden state representation; for Evo 2 models, we use the output logits. To ensure a fair comparison across models with different embedding dimensions, we train a two-layer MLP regression head on top of these features. The hidden layer size of the MLP is adjusted for each model to maintain a consistent number of trainable parameters, with a ReLU activation between layers. We found that using all logit dimensions for Evo 2 models led to training instability. We therefore stabilized training by using only the logits corresponding to the four canonical nucleotides (A, T, C, G). For all models, we use the sequence of token representations as input to the MLP, as this performed comparably to mean-pooling while retaining more information. Each head is trained for 20 epochs using a cosine learning rate scheduler with a 1-epoch warmup, selecting the best learning rate from {3e-5, 1e-4, 3e-4} based on validation performance on round 28 of the fixed future-round split used for model selection.

### C.4. Adaptation Checks on the Fixed Future-Round Split

To assess whether training a larger number of parameters could bridge the future-round ranking gap observed with frozen-encoder probing, we evaluated both full model fine-tuning and prompt tuning as targeted adaptation checks. As shown in Table 7, fine-tuning all model parameters provided no significant improvement over frozen-encoder probing. Similarly, despite an extensive hyperparameter search for prompt tuning (including prompt length, position, and initialization),

*Table 7.* Full fine-tuning checks on the fixed future-round split. Rows report test metrics for checked learning rates after training on rounds 1–27, validating on round 28, and testing on rounds 29–31.

| Model | Learning Rate | Spearman | Recall@10% | nDCG@10% |
|---|---|---|---|---|
| | 3e-5 | 0.0432 | 0.1260 | 0.3094 |
| ESM2-8M | 1e-4 | 0.0553 | 0.1270 | 0.3066 |
| | 3e-4 | 0.0407 | 0.1000 | 0.2594 |
| | 3e-5 | 0.0465 | 0.1136 | 0.2828 |
| ESM2-35M | 1e-4 | 0.0479 | 0.1157 | 0.2817 |
| | 3e-4 | 0.0072 | 0.1033 | 0.2602 |
| | 3e-5 | 0.0271 | 0.1418 | 0.3203 |
| ESM2-150M | 1e-4 | 0.0391 | 0.1127 | 0.2784 |
| | 3e-4 | 0.0271 | 0.1418 | 0.3203 |
| | 3e-5 | 0.0484 | 0.0899 | 0.3119 |
| NT-50M | 1e-4 | 0.0296 | 0.0850 | 0.3055 |
| | 3e-4 | 0.0491 | 0.0848 | 0.3063 |
| | 3e-5 | 0.0367 | 0.0871 | 0.3083 |
| NT-100M | 1e-4 | 0.0460 | 0.0887 | 0.3130 |
| | 3e-4 | 0.0480 | 0.0907 | 0.3106 |
| | 3e-5 | 0.0368 | 0.0841 | 0.3077 |
| NT-250M | 1e-4 | 0.0237 | 0.0840 | 0.3068 |
| | 3e-4 | 0.0485 | 0.0862 | 0.3080 |
| | 3e-5 | 0.0630 | 0.0913 | 0.3147 |
| NT-500M | 1e-4 | 0.0465 | 0.0853 | 0.3064 |
| | 3e-4 | 0.0492 | 0.0888 | 0.3117 |

*Table 8.* Prompt-tuning checks on the fixed future-round split. Rows report test Spearman for prompt initialization, position, length, and layer settings under the same protocol.

| Model | Prompt Init | Prompt Position | Prompt Length | Prompt Layers | Test Spearman |
|---|---|---|---|---|---|
| | kaiming | add | 1 | 0 | 0.0405 |
| | uniform | add | 1 | 0 | 0.0683 |
| | uniform | prepend | 1 | 0 | 0.0617 |
| | uniform | prepend | 2 | 0 | 0.0653 |
| NT-50M | uniform | prepend | 4 | 0 | 0.0652 |
| | uniform | prepend | 8 | 0 | 0.0735 |
| | uniform | prepend | 16 | 0 | 0.0656 |
| | uniform | prepend | 32 | 0 | 0.0754 |
| | uniform | prepend | 64 | 0 | 0.0606 |
| | uniform | prepend | 1 | 0 | 0.0182 |
| ESMC-300M | uniform | prepend | 1 | 10 | 0.0266 |
| | uniform | prepend | 1 | 20 | 0.0239 |
| | uniform | prepend | 1 | 30 | 0.0193 |

performance remained poor, with Spearman correlation failing to exceed $\rho \approx 0.1$, as shown in Table 8. These results suggest that the performance bottleneck is not the number of trainable parameters but rather a more fundamental past-to-future generalization failure of the pre-trained representations on this future-round task under the fixed split.

