# OpenReview forum: "TadA-Bench: A Million-Variant Benchmark for Future-Round Discovery Toward Agentic Protein Engineering"
_ICML.cc/2026/Conference — ICML 2026 regular_

### Official Review · Reviewer_ccy9 · 2026-03-12

**Soundness:** 3
**Presentation:** 3
**Significance:** 3
**Originality:** 3
**Overall Recommendation:** 5
**Confidence:** 3

**Summary:**

This paper introduces TadABench-1M, a large-scale wet lab benchmark to evaluate OOD generalization for protein fitness prediction.

**Compliance With Llm Reviewing Policy:**

Affirmed.

**Final Justification:**

The author provided thorough explanations for my concerns. The community will benefit from such large scale dataset. Hope to see the clarifications in the final version.

**Key Questions For Authors:**

1. Whats the off target ratio given that mutations are directed/planned?
2. How do authors distinguish DNA-level point mutations from NGS technical errors?
3. How do authors track temporal relationships if the graph is built with aggregation and BFS assgined levels?
4. Can the authors show the alignment results between the splits to demonstrate the OOD, like how many sequences change and do they result in a very different structures. Similarly, for those isolated clusters in the t-SNE plot in Appendix C, are there important structural shifts.
5. Why does Evo2 round-based curve start at >10^5 in Figure 5?
6. With up to 25 accumulated mutations in some variants, reduced fitness may arise from protein misfolding/instability rather than reduced catalytic activity. How do the authors account for this potential confounding factor when interpreting activity rankings?

**Limitations:**

Yes. Authors frankly discussed the OOD issue for large base models.

**Strengths And Weaknesses:**

**Soundness**
Both *in silico* and *in vitro* protocals are techinically sound.

**Presentation**
 - "Submission and Formatting Instructions for ICML 2026" change your headers into your actual title.
 - Consistently italicize "***in silico***" and "***in vitro***"

**Significance** This paper provided new benchmark dataset for deep experimental rounds.

**Originality** Provided original dataset and effective framework for computing correlation.

**Strength** Nice open-sourced code preparation; OOD naturally occur within later rounds in directed evolution experiments

**Weakness** See questions

---

> ### Author Rebuttal · Authors · 2026-03-28
>
> Thank you for the constructive comments and for the positive assessment of the benchmark. We will also fix the presentation issues you noted.
>
> 1. Formatting. We will replace the running header with the paper title, consistently italicize in silico / in vitro, and fix the missing y-axis information in Fig. 5.
>
> 2. Off-target/unplanned mutations. We likely need to distinguish between two different interpretations of this question more clearly. If "off-target" refers to genome-wide off-target editing of a deployed base editor, that is outside the scope of TadABench, which measures selection-coupled fitness of TadA variants rather than cellular off-target editing profiles. If it refers to variants outside the intended library design, we do not currently report a quantitative designed-vs-observed ratio in the paper, and we will clarify this more carefully. The benchmark is sequence-defined rather than design-intent-defined: final labels are assigned to NGS-observed variants rather than to an expected mutation list.
>
> 3. Distinguishing true mutations from NGS technical errors. We mitigate this in several ways: very deep sequencing reduces sampling noise; QC and read-support filtering remove low-confidence observations; Seq2Graph relies on local within-round relative comparisons rather than fragile absolute counts; and cross-round integration is supported by repeated sequence overlap. In addition, the supplement's subsampling/robustness analyses show that the resulting labels are stable under substantial resampling. We will make these safeguards more explicit in the revision.
>
> 4. Temporal relationships vs. BFS levels. Temporal information comes from the recorded metadata of the experimental rounds, not from BFS levels. The graph is used for label unification across rounds. BFS / shortest-path propagation is only a computational step for assigning relative activity scores on the cleaned DAG, and should not be interpreted as temporal ancestry.
>
> 5. Demonstrating the OOD shift more explicitly. We agree and will foreground the existing analyses more clearly. The appendix already contains a t-SNE with round-wise clustering and an inter-round similarity matrix showing that some rounds occupy different regions of sequence space. Importantly, we do not claim wholesale fold changes; the challenge is a realistic intra-family distribution shift within the same TadA scaffold. The isolated clusters are therefore better interpreted as distinct library branches or mutational subspaces rather than as evidence of fundamentally different structures.
>
> 6. Why does the Evo2 round-based curve start above 1e5 in Fig. 5? Because the round-based setting selects whole experimental rounds rather than arbitrary subsamples. The smallest selected round bundle already contains on the order of 1e5 sequences, so that the curve naturally starts later on the x-axis.
>
> 7. Misfolding/instability as a confounder. We appreciate the reviewer raising this important distinction. In our study, the "activity" we measure represents the final, end-to-end functional output of the variant in a cellular context. For a variant to exhibit measurable editing activity, it must successfully navigate the entire biological pipeline: transcription, expression, stable structural folding, and finally, specific base editing. Therefore, misfolding or instability are not "confounders" in our assay; rather, they are intrinsic biological bottlenecks that naturally dictate the overall functional fitness we aim to measure. We will explicitly clarify in the manuscript that our rankings reflect this comprehensive in vivo performance rather than an isolated catalytic constant. Furthermore, our orthogonal GFP assay validates that this holistic functional ranking is biologically meaningful.

---

> > ### Author Rebuttal · Reviewer_ccy9 · 2026-04-03
> >
> > The author provided thorough explanations for my concerns. The community will benefit from such large scale dataset. Hope to see the clarifications in the final version. I will raise my score.

---

### Official Review · Reviewer_K3NW · 2026-03-12

**Soundness:** 3
**Presentation:** 3
**Significance:** 3
**Originality:** 3
**Overall Recommendation:** 4
**Confidence:** 3

**Summary:**

Protein engineering iteratively searches for high-affinity binders through multiple rounds of sequence editing. This paper focuses on the prokaryotic deaminase enzyme TadA, allowing us to estimate the evolutionary trajectory of a sequence over multiple rounds and provide benchmarks for enzyme activity measured at each round. Enzyme activity is measured by the abundance of TadA variants in the population pool of each round, with higher enzymatic activity replicating more efficiently leading to their enrichment. The activity of a TadA variant is proportional to its replication rate, which is measured by read counts. To align benchmark results with real-world protein engineering, we propose a temporal split that can also operate on unseen proteins (OOD scenario).

**Compliance With Llm Reviewing Policy:**

Affirmed.

**Key Questions For Authors:**

[Major]
- How can we demonstrate that temporal splitting is a valid OOD setup, and how different are the resulting data distributions
- When applying a sequence mutation, is editing performed one at a time? Are there cases where editing occurs at multiple positions?
- If activity assignment is done using a DAG after multiple rounds of pooling, how do you infer the evolutionary path of each sequence?
- Is benchmarking possible for other OOD scenarios (e.g. cell-line split, structure similarity split)?

**Limitations:**

Yes

**Strengths And Weaknesses:**

[Soundness]
- Strict measurement consistency guaranteed by standardized wet-lab experiments is strength of this benchmark.
- As I understand it, relative activity ranking evaluates activity by measuring how much the replicate count increases compared to the previous round after introducing a mutation. If sequences from multiple rounds of mutations are pooled together and activity is assigned using a DAG, then an explanation is needed for how the evolutionary path of each sequence is inferred.

[Presentation]
- There's a lack of demonstrations that shows temporal splitting is truly a good OOD setting. It should show how different the distributions are.

[Significance]
- Due to the lack of consensus of benchmark datasets in biomedical domain, benchmark dataset release is valuable that is truely valuable assets for community.
- Split is limited to temporal splits. It is insufficient to claim that this can be used for rigorous OOD generalization.

[Originality]
- Benchmark dataset seems original because it gives the measurements for multiple rounds. For protein-protein binding affinity prediction, there is a dataset that measures the difference in binding affinity before and after mutation, but as far as I know, there is no one that measures activity for multiple rounds for a single protein.

---

> ### Author Rebuttal · Authors · 2026-03-28
>
> Thank you for the constructive review and for highlighting the value of releasing a benchmark in this area. We clarify the main technical points below.
>
> 1. Why is the temporal split a valid OOD setting? Our intended claim is specifically temporal OOD / future-round generalization in directed evolution. We believe this is a meaningful OOD setting for three reasons: (i) it matches the real use case of predicting future experimental rounds from past rounds; (ii) the shift is empirically visible, since the same benchmark is highly learnable under a random split (Spearman ~0.8) but collapses under the temporal split (~0.1); and (iii) the appendix already shows round-wise clustering in t-SNE space and substantial inter-round variation in the sequence-similarity matrix. We will surface this evidence more prominently and avoid implying that the temporal split exhausts all OOD notions.
>
> 2. Are mutations introduced one at a time? No. The libraries are generated by expert-designed degenerate synthesis and can contain multiple simultaneous mutations. Many variants are therefore combinatorial mutants rather than single-step edits, and some sequences accumulate more than 25 amino-acid changes.
>
> 3. How is the evolutionary path inferred after pooling multiple rounds? We agree this distinction needs clarification. Seq2Graph does not attempt to infer a biological lineage from the graph structure. Rather, the evolutionary path is empirically determined because we retain the round information for each sequence; the temporal succession of these rounds dictates the evolutionary order. The DAG's role is separate: it links rounds through exact sequence overlap to establish a relative scoring system. After removing cycles caused by noise, BFS/shortest-path propagation is used solely to assign globally consistent activity scores from a reference sequence. We will add a section clarifying that graph distances represent computational score propagation, not temporal ancestry.
>
> 4. Other OOD scenarios. We agree that temporal split is one important OOD axis, not the only one. Mutation-distance or structure-similarity splits could, in principle, be defined on the same dataset, while a cell-line split would require additional matched multi-context assays that are not part of the current data. We will revise the wording so the paper presents TadABench as a rigorous temporal/evolutionary OOD benchmark supported by the current dataset, rather than as an exhaustive treatment of all OOD settings.

---

> > ### Author Rebuttal · Reviewer_K3NW · 2026-04-03
> >
> > Thank you for your response. By narrowing the scope of OOD, the objective of the benchmark has become clearer, and I believe it can be considered a rigorous benchmark.

---

### Official Review · Reviewer_sUJ6 · 2026-03-13

**Soundness:** 3
**Presentation:** 4
**Significance:** 3
**Originality:** 4
**Overall Recommendation:** 5
**Confidence:** 4

**Summary:**

The authors introduce TadABench, a large dataset of TadA enzyme variants generated through 31 rounds of directed evolution for evaluating protein foundation models in a realistic experimental optimization setup with temporal cutoffs. They benchmark a representative set of protein and genome language models on this task. Overall this is a welcome contribution to the field and quantifies key concerns about the utility of PLMs and GLMs for experimental design.

**Compliance With Llm Reviewing Policy:**

Affirmed.

**Final Justification:**

I strongly support the publication of this benchmark. It is a good compliment to the ProteinGym benchmark in an orthogonal setting, and I expect many methods have over-indexed on the former and this seems to be a good course-correction in a more useful setting. I recommend authors take the time to set up public contributing instructions and leaderboards for continuous evaluations.

**Key Questions For Authors:**

Are there any insights about what would drive performance improvement? Results seem to signal that pretraining size and modality are not particularly important for this task.

Nit: Fig 5 y axis is missing.

**Limitations:**

yes

**Strengths And Weaknesses:**

Strengths
1. The benchmark dataset is unusually large, experimentally grounded, and deep, containing over one million variants derived from 31 rounds of directed evolution.
2. The temporal split design reflects realistic protein engineering workflows where future mutations must be predicted from past rounds.
3. The empirical comparison across several protein language models highlights a clear gap between iid evaluation and generalizing to future experiments.
4. The Seq2Graph framework proposes an algorithmic method for integrating multi-round seq2func experimental measurements and correcting batch effects.

Weaknesses
1. The primary contribution is a dataset and evaluation protocol. The algorithmic novelty of Seq2Graph is limited because it relies on known components such as ranking graphs and feedback arc set heuristics.
2. The central claim that evolutionary depth rather than data volume drives OOD generalization does not seem well substantiated. Supported only by limited ablation experiments and lacks statistical analysis.
3. The evaluation excludes strong structure-aware or supervised protein fitness models that could provide stronger baselines.
4. The benchmark focuses on a single enzyme family (TadA),  raising concerns about generalizability to broader protein design tasks.

---

> ### Author Rebuttal · Authors · 2026-03-28
>
> Thank you for the constructive feedback and strong overall support. We agree that this is primarily a benchmark paper, and we will revise the framing accordingly.
>
> 1. Seq2Graph novelty. We agree that the benchmark and evaluation protocol are the main contribution. Seq2Graph is not intended to be presented as a fundamentally new graph-theoretic primitive; its role is to serve as the enabling integration method that turns 31 rounds of noisy enrichment measurements into a single consistent million-scale fitness landscape. We will revise the wording to avoid overstating algorithmic novelty.
>
> 2. "Evolutionary depth rather than data volume" claim. We agree this needs to be stated more carefully. Our intended claim is an empirical finding on this benchmark, not a universal causal law. In the scaling study, diversity- and round-based curation consistently outperform density-based subsampling at matched training sizes, suggesting that coverage of distinct evolutionary regions is more informative for future-round prediction than simply increasing local density from already-seen rounds. We will soften the wording accordingly and, where possible, add or discuss statistical variability more explicitly.
>
> 3. Stronger baselines. This is a fair suggestion. Our goal in the current paper was to compare general-purpose pretrained biological foundation models under a shared lightweight protocol, rather than to claim an exhaustive leaderboard over all possible specialized fitness predictors.
>
> 4. Single-family scope. We agree that TadABench should not be interpreted as a universal protein-design benchmark. Our intended positioning is a high-fidelity benchmark for temporal OOD in one therapeutically relevant family, precisely because that allows unusual evolutionary depth and assay consistency without cross-study batch effects. We will narrow the wording accordingly and present TadABench as complementary to broad multi-family benchmarks rather than a substitute for them. The Cas9 appendix is included only to show that the construction pipeline itself is not intrinsically TadA-specific.
>
> 5. What may improve performance? The current results suggest that larger generic pretraining or modality changes alone are insufficient once evaluation moves forward along an evolutionary trajectory. Promising directions include stronger modeling of epistasis, ranking / selection-aware objectives, trajectory-aware adaptation, and broader experimentally grounded coverage of functionally distinct regions. We will add this interpretation more clearly while keeping it as a hypothesis suggested by the benchmark rather than a claim proven by the current experiments.
>
> 6. Fig. 5. Thank you for catching this. We will add the missing y-axis information in the revision.

---

> > ### Author Rebuttal · Reviewer_sUJ6 · 2026-04-02
> >
> > Thank you for your response. I strongly support the publication of this benchmark. It is a good compliment to the ProteinGym benchmark in an orthogonal setting, and I expect many methods have over-indexed on the former and this seems to be a good course-correction in a more useful setting. I recommend authors take the time to set up public contributing instructions and leaderboards for continuous evaluations.

---

### Official Review · Reviewer_AwCC · 2026-03-14

**Soundness:** 3
**Presentation:** 3
**Significance:** 3
**Originality:** 3
**Overall Recommendation:** 5
**Confidence:** 4

**Summary:**

The authors propose a new benchmark dataset of TadA enzyme DNA sequences that have undergone multiple rounds of evolution. 14 existing biological language models (for DNA or protein) are evaluated on the novel benchmark dataset. The authors design three types of evaluations to quantify generalizability: density, diversity, and round. The benchmark dataset is validated via an orthogonal in vitro experiment and an in silico analysis.

**Compliance With Llm Reviewing Policy:**

Affirmed.

**Final Justification:**

The authors have addressed all of my concerns.

**Key Questions For Authors:**

Are there protein benchmarks that contain the TadA or Cas9 protein? If so, could you compare your dataset against the existing ones? How similar/different are the variants observed in these existing protein benchmarks versus yours (e.g., diversity)? Is your benchmark complementary or overlapping with existing protein benchmarks?

How do the existing biological language models perform on the Cas9 dataset that you have constructed? Are the observed trends regarding generalizability consistent?

How different are the sequences in each round? From Figure 8, it seems like there is clustering of sequences by the round of protein evolution as well as clusters of round-clusters. Is it necessarily the case that more rounds result in larger difference from round 1 (as in, proteins that have undergone round 31 look the most different from proteins that have only undergone 1 round)?

Are all of the sequences, including those from intermediate rounds, functional?

I don't see any link to the dataset. How will the dataset be released if the paper is accepted?

**Limitations:**

Yes

**Strengths And Weaknesses:**

Strengths
- This seems like a very valuable resource.
- 14 existing biological language models (for DNA or protein) are evaluated on the novel benchmark dataset
- The authors design three types of evaluations to quantify generalizability: density, diversity, and round.
- The benchmark dataset is validated via an orthogonal in vitro experiment and an in silico analysis.

Limitations
- See questions below

---

> ### Author Rebuttal · Authors · 2026-03-28
>
> Thank you for the positive assessment and for highlighting the benchmark's value. We respond point-by-point.
>
> 1. Relation to existing TadA/Cas9 benchmarks. Broad benchmark suites provide breadth across many proteins and assays. TadABench is intended to be complementary rather than redundant: to the best of our knowledge, there is no public TadA resource with comparable 31-round depth, standardized wet-lab consistency, unified cross-round labels, and a forward-in-time split. Even where some canonical backbones or individual variants may overlap with prior studies, the benchmark setup, label construction, and generalization task are materially different. We will clarify this positioning more explicitly in the revision.
>
> 2. Cas9 results. In the current paper, the Cas9 dataset is included only to show that Seq2Graph is not TadA-specific as a construction framework. We did not intend to present Cas9 as a second fully benchmarked biological conclusion without showing the corresponding model study. We will clarify this scope and restrict the main empirical claims to TadA in the current submission. A full Cas9 benchmark would be valuable future work.
>
> 3. How different are rounds, and does later always mean farther from round 1? Not necessarily. The shift is real, but it is not monotonic in the round index. The t-SNE and the round-wise similarity matrix show clear clustering by round as well as higher-order "clusters of rounds," consistent with guided and partially branching exploration of sequence space. Therefore, later rounds are not assumed to be strictly farther from round 1 than every intermediate round. We will make this point clearer in the text and captions.
>
> 4. Are all sequences functional? Not in a binary sense. The benchmark is intentionally continuous / ranking-based and spans a wide range from weak to strong variants. Presence in the screened population does not imply that a sequence passes some independent functional threshold; rather, it receives a relative fitness score under the selection system. We will clarify this label interpretation more explicitly.
>
> 5. Release plan. Please note: the dataset is included in the current attachments for review. We agree this should have been stated more clearly. Upon acceptance, we plan to release the processed benchmark on HuggingFace. We will make this release plan explicit in the revision.

---

> > ### Author Rebuttal · Reviewer_AwCC · 2026-04-03
> >
> > Thank you for addressing my questions! I will raise my score +1.

---

### Decision · Program_Chairs · 2026-04-30

**Decision:**

Accept (regular)

**Comment:**

This paper proposes a new large-scale wet-lab protein benchmark and the reviewers unanimously support acceptance for its contribution to the community as it is of substantial scale and expected to benefit future research. The reviewers left suggestions, such as ensuring clear documentation, contribution guidelines, and a public leaderboard for evaluation, are suggested to be addressed in the final version.